# Mechanical Properties of Graphene Networks under Compression: A Molecular Dynamics Simulation

**DOI:** 10.3390/ijms24076691

**Published:** 2023-04-03

**Authors:** Polina V. Polyakova, Julia A. Baimova

**Affiliations:** Institute for Metals Superplasticity Problems of RAS, Khalturina St., 39, 450001 Ufa, Russia; polina.polyakowa@yandex.ru

**Keywords:** multilayer graphene, mechanical properties, molecular dynamics, deformation

## Abstract

Molecular dynamics simulation is used to study and compare the mechanical properties obtained from compression and tension numerical tests of multilayered graphene with an increased interlayer distance. The multilayer graphene with an interlayer distance two-times larger than in graphite is studied first under biaxial compression and then under uniaxial tension along three different axes. The mechanical properties, e.g., the tensile strength and ductility as well as the deformation characteristics due to graphene layer stacking, are studied. The results show that the mechanical properties along different directions are significantly distinguished. Two competitive mechanisms are found both for the compression and tension of multilayer graphene—the crumpling of graphene layers increases the stresses, while the sliding of graphene layers through the surface-to-surface connection lowers it. Multilayer graphene after biaxial compression can sustain high tensile stresses combined with high plasticity. The main outcome of the study of such complex architecture is an important step towards the design of advanced carbon nanomaterials with improved mechanical properties.

## 1. Introduction

Graphene, a two-dimensional (2D) structure with excellent mechanical and physical properties, is considered one of the most promising materials (along with the three-dimensional (3D) derivatives) for electronics, spintronics and energy storage [1,2,3,4]. To date, several methods for obtaining mono- and multilayer graphene are known: the mechanical and chemical exfoliation of graphite [5,6]; the epitaxial growth of graphene on silicon carbide [7], titanium carbide [8] and different metal substrates, such as Ni, Cu, Pt and Co [9,10]; organic synthesis [11]; chemical vapor deposition [12]; and the synthesis of graphene from graphite oxide [13].

Although graphene has unique properties and great potential, there still exist many problems in the process of graphene fabrication and application. From this point of view, the assembling of 2D graphene into a 3D structure with complex architecture can become a green fabrication of new structures with good mechanical and physical properties. In such a way, different graphene networks (fibers, aerogels and crumpled graphenes) can be obtained with different types of hybridization and properties that are considerably dependent on the morphology.

Based on the methods for graphene fabrication, synthesis methods for the fabrication of 3D graphene with complex architecture have been developed, such as chemical vapor deposition, a reduction from graphene oxide and exfoliation from the liquid phase [14]. Graphene networks (aerogels and fibers) are the lightest materials with high specific surface area, commonly with high strength and exceptional physical properties, such as high conductivity and thermal stability.

These materials are shown to have very good elasticity, a tunable Poisson’s ratio [15], irreversible damage [16] and high strength [17,18]. It was also found that such structures can be applied as adsorbents or energy storage devices since they have a high specific surface area and compression recoverability [19,20]. In addition, many of these materials have been revealed to have super compressive properties, i.e., compressed to large strain without damage and can recover after release, which might be used as flexible conductors or highly sensitive strain sensors [21,22].

Due to its porous nature, graphene networks can be excellent candidates for the storage and adsorption of other elements; for example, hydrogen or catalytic metal nanoparticles [23]. Graphene aerogels also show a wide bandwidth, high electrical conductivity [24], thermal insulation [25] and high compressibility [23,26,27]. The fabrication methods used to obtain various 3D graphenes will significantly affect their physical and mechanical properties [28]. Such 3D frameworks have excellent adsorption properties, particularly in the presence of metal nanoparticles supported on graphene [29,30,31]. Thus, such 3D structures demonstrate high potential in catalysis, energy storage, actuators and environmental protection [32].

Despite different studies on mechanical properties having been conducted, important issues, such as the mechanical behavior under different treatments, the effect of impurities and the structural morphology should be understood better. Molecular dynamics (MD) simulation is an effective tool for the investigation at the atomistic level of mechanical behavior and physical properties of various nanostructures subjected to external loading [33,34,35,36,37,38].

The graphene network is also under rigorous attention; thus, numerous studies have been conducted using MD simulations [17,18,22,39,40,41]. In such studies, a graphene network is composed of single-layer or multi-layer graphene interconnected by both van der Waals and covalent bonds, with all types of hybridization from sp to $sp^{3}$, since graphene flakes in this network are packed together by chemical bonds, physical adhesion and structural constraints [18,40,42].

The nature of graphene itself results in the crumpling, bending and buckling of graphene flakes and the formation of curved graphene structures. Graphene networks can either easily transform to instability or demonstrate deformation recovery under compression [18,43,44,45,46]. Assembling graphene into various 3D networks with new physical and mechanical properties also attracts considerable attention. For such graphene networks, the morphology under consideration can significantly affect the mechanical properties. Thus, the understanding of the relationship between the mechanical (physical) properties and structure peculiarities is of high importance.

In the present work, MD simulations are applied to multilayered graphene compressed to a graphene network with special morphology. The structural and mechanical properties of the obtained graphene network are investigated under tension.

## 2. Results and Discussion

### 2.1. Multilayer Graphene under Compression

In Figure 1c, pressure–strain curves (black solid line) and in Figure 1d, potential energy–strain curves (blue solid line) are presented for MG under compression. Snapshots of the whole structure and one graphene layer are also presented in Figure 1e. The energy–strain curve coincides well with the stress–strain curve, which means that the stress–strain behavior of materials can also be interpreted by the energy analysis. Very similar structures that are highly dense and yet nano-porous were studied previously both in experiments and simulations [41,47]. The curved graphene layers attracted each other and assembled into uniform and compact networks.

It was shown previously that graphene networks can be super compressive (compressed to large strain without damage), which is in good agreement with the obtained results [48]. The pressure–strain curves obtained experimentally qualitatively coincide with the compression curve for pure MG in Figure 1c. The values of the achieved compression are much higher, which is explained by the limitations of the MD simulations. While this method can be very carefully used for the quantitative description of deformation properties, it can also successfully qualitatively describe the mechanical behavior.

As it can be seen, until $\varepsilon_{cp}$ = 0.21, the pressure in the system was close to zero (stress plateau) since, at this deformation stage, almost no interaction between separate graphene layers took place and free volume was removed. Corrugation (rippling) of graphene can be seen even at the first deformation stages (for example, structure A). However, with the increase in compressive strain, the edges and corners of folds became sharp (structure B). From the energy–strain curve, it is also seen that there are no changes in the potential energy until $\varepsilon_{cp}$ = 0.21. After $\varepsilon_{cp}$ = 0.21, a slight increase in the pressure in the system and a small jump on the energy–strain curve can be seen. This is connected with the slippage of some graphene layers to each other.

The change in the course of the curves at the point $\varepsilon_{cp}$ = 0.3 is associated with an interaction of graphene layers, which moved closer to each other (a densification process). Moreover, the appearance of a considerable number of sharp edges also contributes to the increase in the potential energy of the system. Approximately up to $\varepsilon_{cp}$ = 0.3, graphene has a rippled structure (Figure 1e) with an interlayer distance 3.4 Å. After compression, to $\varepsilon_{cp}$ = 0.43, all graphene layers have sharp edges (Figure 1a, structural element). For comparison, the compression of one graphene layer in the simulation cell of the same size ($L_{x}^{0}$ = $L_{y}^{0}≃$ 100 Å, $L_{z}^{0}≃$ 47 Å) was also performed. We found that sharp corners and folds were not formed without the presence of other graphene layers.

Thus, after compression, a special type of multilayered graphene fiber was obtained. The MG was composed of strongly crumpled graphene layers with an interlayer distance of 3.4 Å and a density of 3.0 g/m${}^{3}$. From the common neighbor analysis, we found that there was no bonding between graphene layers (MG has $sp^{2}$-hybridisation). From the pressure–strain curve, it is seen that this structure behaves similar to a sort of crumpled graphene or paper balls: the increase in stiffness and strength under an applied compression can be explained by the formation of more hard-to-bend ridges [48,49,50,51,52,53]. Therefore, the high bending stiffness of multilayer graphene is maintained.

### 2.2. Multilayer Graphene with Ni and H Nanoclusters

Let us discuss the compression process in the presence of other molecules and nanoclusters, such as hydrogen or metal. In Figure 2, snapshots of a part of the simulation cell for MG with Ni and MG with H in the initial state (a), during relaxation (b) and compression (c) are presented. Pure MG and MG with H nanoclusters behave similarly since, during relaxation, H clusters transform to separate H${}_{2}$ molecules and H atoms, which spread over graphene (Figure 2b). Thus, during relaxation, MG with increased distance transforms to MG with 3.34 Å. Moreover, during compression, these separate H atoms and molecules cannot considerably affect the process of compression. Thus, the pressure–strain and energy–strain curves are not presented in Figure 1c,d.

In comparison, MG with Ni nanoclusters behaves differently during relaxation. For MG with Ni nanoclusters, the distance between graphene layers even increased during relaxation from 7.85 to 9.15 Å (Figure 2b). The chosen diameter of Ni nanoclusters allows obtaining this special MG with the given interlayer distance. Thus, by varying the size of a metal nanocluster, one can obtain a specific distance between graphene layers in MG.

Then, compression was performed for a pure MG and MG with Ni. For comparison, both pressure–strain and potential energy–strain curves are presented in Figure 1c,d for pure MG (black and blue solid lines) and MG with Ni nanoclusters (green and orange dashed lines). The curves for both structures coincide well with the small differences in the achieved compressive strain. This increase is explained by the greater achieved density ρ = 3.3 g/m${}^{3}$ due to the presence of Ni nanoclusters. The slow decrease in the potential energy (orange dashed curve) is explained by the fact that Ni nanoclusters are strongly attracted by graphene layers, which are energetically more favorable. The Ni nanoclusters are attracted by graphene layers from both sides and bond graphene layers together.

In Figure 2c, a snapshot of a part of the MG with Ni during biaxial compression is shown. It can be seen that the Ni nanoclusters can change their shape: Ni atoms can spread between two graphene layers, while others preserve their initial shape. Nickel nanoclusters are located on folds and are uniformly distributed between layers.

### 2.3. Mechanical Behavior of Multilayer Graphene under Uniaxial Tension

In Figure 3, the stress–strain curves for MG during tension are presented. Uniaxial tension along the *x*-, *y*- and *z*-axes is achieved to analyze the anisotropy of obtained MG. Stress σ (ε) is $\sigma_{xx}$ ($\varepsilon_{xx}$) for tension along the *x*-axis, $\sigma_{yy}$ ($\varepsilon_{yy}$) for tension along the *y*-axis and $\sigma_{zz}$ ($\varepsilon_{zz}$) for tension along the *z*-axis.

The value of strain is calculated at each timestep as $\varepsilon_{xx}\;=\;\left(L_{x}^{final}\;-\;L_{x}\right)/L_{x}$; $\varepsilon_{yy}=\left(L_{y}^{final}-L_{y}\right)/L_{y}$ and $\varepsilon_{zz}=\left(L_{z}^{final}-L_{z}\right)/L_{z}$. Here, for simplicity, only pure MG is considered. It can be seen that there is a strong connection between the stacking of graphene layers and the mechanical properties of MG. In comparison with the graphene networks with random orientation [41], there is no elastic regime, which is explained by the considered graphene stacking.

In Figure 4, Figure 5 and Figure 6, corresponding snapshots of MG and a single graphene layer during tension along the *x*-, *y*- and *z*-axes, correspondingly, are presented for some critical points shown in Figure 3.

Let us discuss the tensile loading along the *x*-axis (red curve in Figure 3). Up to ε = 0.6, the stress is equal to zero, which is explained by the straightening of graphene layers along the *x*-axis (see Figure 4a,c(II’)), with simultaneous changes of the shape of graphene in the yz-plane. To understand the stress–strain behavior, in Figure 4, snapshots of the MG during tension are presented in the (a) xy-plane and (b) yz-plane, together with a single graphene layer shown in a perspective view (c). As can be seen from Figure 4a,c(initial), there are folds with rigid corners oriented along the loading direction (the opening angle is shown by red arrows).

In Figure 4b,c(II’), for clearance, the edge of a single graphene layer taken from the center of the simulation cell is shown by the red dotted line. This means that, at about ε = 0.63, each graphene layer in the MG graphene transforms from being folded in different directions to graphene with one fold (flattened) along other directions. After that, the stress increase is defined by the stretching of the flattened graphene layer. It should be also noted that, during tension along the *x*-axis, an increase in bond length in the basal plane of graphene also took place.

Further, the stress increase is explained by the slippage of graphene layers and the rearrangement of atoms with the formation of new covalent bonds between neighboring layers. The appearance of new $sp^{3}$ bonds results in the increase in tensile strength. The crystalline order retains up to ε = 0.73, after which the transformation to an amorphous carbon structure is observed. Pop-in events on the stress–strain curve along the *x*-axis (red curve, Figure 3) are associated with the formation of nanopores. However, for amorphous carbon continuous rearrangement of the atomic structure is observed: some bonds between atoms break, and new bonds appear during tension. The ultimate tensile strength for tension along the *x*-axis is 172 GPa with a fracture strain of 1.0.

The tensile loading along the *y*-axis (blue curve in Figure 3) is similar to the tension along the *x*-axis. In Figure 5, snapshots of the MG during tension are presented in (a) the xy-plane together with a single graphene layer shown in a perspective view (b). The tension direction is shown by black arrows. Here, the yz-plane is not presented, since no changes can be seen from the yz-projection, and folds cannot be visually distinguished.

During tension along the *y*-axis, the increase in bond length in the basal plane of graphene took place until ε = 0.35. Again, there was a transition from sharp folds to soft folds (from point I” to point II” in Figure 3). In comparison to tension along the *x*-axis, the folds did not appear during tension, they only became softer (Figure 5b(II”)). The appearance of new ripples on the sides of initial folds was observed (blue circle in Figure 5b(II”)). Pop-in events on the tension curve along the *y*-axis (blue curve, Figure 3) are associated with the formation of nanopores but without immediate fracture. The ultimate tensile strength for tension along the *y*-axis was 144 GPa with a fracture strain of 1.18.

The highest fracture strain was observed for tension along the *z*-axis (see Figure 3, black curve). The fracture strain and ultimate tensile strength for tension along the *z*-axis were 1.97 and 108 GPa, respectively. Very similar mechanical behavior of graphene networks under tension was shown in [41,54,55,56]. In Figure 6, snapshots of two graphene layers from the center of MG are presented during tension along the *z*-axis. The MG sample is not presented, since the structure transformation can be better seen from the changes for two layers.

During biaxial compression, the size of the simulation cell was not changed along the *z*-axis, while it was reduced along the *x*- and *y*-axis. At these conditions, graphene layers are elongated along the *z*-axis but compressed along the *x*- and *y*-axes. From Figure 3 (black solid line), it is seen that, up to ε = 0.56, the tensile stress is close to zero (or even lower), almost not changed since the graphene layers are slightly rearranged along the *z*-axis (moving in the empty space of the simulation cell). After compression, the equilibrium angle is equal to 45${}^{\circ }$, and it is decreased up to 32${}^{\circ }$ (a point I in Figure 6) at ε = 0.56. Graphene layers became more corrugated and more compressed but still can easily slide over each other, which allows further deformation.

At ε = 0.56, a slight stress increase took place, which is explained by the rearrangement of graphene layers (Figure 6, point II). As shown in Figure 6, the flexure angle of graphene layers changes from 32${}^{\circ }$ to 16${}^{\circ }$ at a tensile strain of about 0.75 and remains unchanged during further deformation. At the previous step of deformation, changes in the flexure angles were easier since there was empty space for movement of the sides of graphene layers.

Here, neighboring graphene layers become much closer to each other, and a further decrease in flexure angles is complicated. However, at ε = 0.56, the slippage of graphene layers over each other took place to adapt to the external loading. Surface-to-surface contact is the main contact style for MG networks, which contributes to their plastic deformation, and graphene slippage mainly occurs in the plastic deformation stage [41].

At ε = 0.75 (point III), a stress decrease took place again since the new structural state of graphene layers was observed: the top layer of graphene (shown by gray color in Figure 6) separated from the bottom layer (shown by light red in Figure 6), and the walls of each layer began to contact each other (shown by a black circle in the figure). Until ε = 1.0 (point IV), all six layers were rearranged to the new state, which was accompanied by the stress decrease. Other changes in the course of the stress–strain curve (Figure 3, black curve) are associated with the further rearrangement of graphene layers (Figure 6, points IV–VI) due to the breaking of van-der-Waals bonds inside the structure and the formation of new bonds.

As mentioned previously [57], the interlayer distance between graphene layers is of high importance for the mechanical properties of such structures. The smaller the interlayer distance, the higher the mechanical properties. Moreover, the compactness of the structure can be controlled during its fabrication. In our work, during tension along the *x*-axis, the layers were straightened along the tension direction with a simultaneous decrease in the interlayer distance between graphene from 3.4 Å to 2.7–3.0 Å.

For the tension along the *y*-axis, a decrease in the interlayer distance between sheets from 3.4 Å to 2.8–3.1 Åwas also observed. However, during tension along the z-direction, no such changes were observed, which can affect the slipping of graphene layers over each other. For such layered graphene networks, one of the main deformation mechanisms is the sliding of graphene layers over each other, since the van-der-Waals bonding is quite weak [41].

In [47], the mechanical properties of nano-porous graphene networks with high density were studied. The obtained ultimate tensile strength differs from our results quantitatively, which is explained by the high strain rate applied in MD simulations (a limitation of the method itself). Moreover, it is complicated to conduct a quantitative comparison of simulation and experimental works, since the structural parameters can be considerably different.

For example, the pore size in the experimentally synthesized samples is larger than in computational models. The presence of other atoms (such as O, H and N) or defect nucleation will considerably affect the properties, while, in the present simulation, a pure endless graphene network is considered. Nevertheless, qualitatively very similar results are obtained, and such molecular dynamics simulation can shed light on the understanding of the mechanical behavior of graphene aerogels/networks.

We found that the mechanical properties decreased as the number of layers increased from monolayer to multi-layer graphene. It is clear that the fracture strain and ultimate tensile strength are not significantly affected by the transition from few-layer graphene to multilayer graphene [58]. For six-layer graphene, it was found that the ultimate tensile strength was 115.6 GPa, and the fracture strain was 0.136 [58]. The difference in strength values can be explained by the absence of defects in the structure in the MD model.

All the obtained graphene networks [41,47,48] have their unique micro-structure that differs by the size of the structural elements and by their placement. The simulated MG is more ideal and has fewer defects than the experimental samples, which can considerably affect the values of the mechanical properties.

## 3. Methods and Materials

In Figure 1a, the initial structure of multilayer graphene (MG) consists of six graphene layers with 3936 carbon atoms each (total number of atoms *N* = 23,616). The size of the initial structure along three dimensions is $L_{x}^{0}$ = $L_{y}^{0}≃$ 100 Å, $L_{z}^{0}≃$ 47 Å. The interlayer distance is *h* = 7.85 Å, which is two-times larger than for graphite ($h_{graphite}$ = 3.34 Å). The initial distance between graphene layers is not typical for MG and is chosen with the aim to construct a 3D carbon structure with the new properties.

In a real experiment, a chosen MG with the interlayer distance increased to 7.85 Å will be unstable as it tends to become MG with an interlayer distance 3.34 Å [58,59]. However, in real experiments, other clusters and molecules can be absorbed by graphene, which can affect the interlayer distance. Nickel is one of the most widespread metals for a range of graphene applications, including to grow carbon nanotubes and graphene [60,61,62], to form a 3D composite structure [63,64,65,66] and to synthesize large-area graphene on a metal substrate [67,68,69]. Thus, three different cases are considered to study the effect of adsorbed molecules on the formation of MG with an increased interlayer distance: (i) pure MG, (ii) MG with Ni nanoclusters between graphene layers and (iii) hydrogen nanoclusters between graphene layers.

The size of one nanocluster is 5 Å. Three nanoclusters are randomly distributed between each layer, and the total number is 18 nanoclusters. In Figure 1a, only pure MG is presented.

Periodic boundary conditions were applied along the *x*-, *y*- and *z*-axes. All the simulations were conducted using the LAMMPS package with the AIREBO [70] interatomic potential for C–C and C–H, which include both covalent bonds in the basal plane of graphene and van-der-Waals interactions between graphene layers. For the case when Ni nanoclusters are added, the Morse interatomic potential is used to describe C–Ni and Ni–Ni interactions [66,71,72,73,74]. The Nose–Hoover thermostat is used to control the system temperature around 300 K. Initially, the structure of the MG is relaxed to achieve the state with the minimum potential energy.

To obtain graphene fiber, biaxial compression is applied ($\varepsilon_{xx}=\varepsilon_{yy}=\varepsilon_{cp}$, $\varepsilon_{zz}$ = 0) at 300 K with a given strain rate $\dot{\varepsilon }$ = 0.01 ps${}^{-1}$. As mentioned in [75], there can be different ways to apply the deformation to the simulation box. In this work, compression is applied to the whole simulation box. During biaxial compression, the volume of the simulation cell is not changed along the *z* direction. The resulting pressure in the system is calculated as $p=(\sigma_{xx}+\sigma_{yy}+\sigma_{zz})/3$.

The size of the structure after compression is $L_{x}$ = $L_{y}≃$ 57 Å, $L_{z}≃$ 47 Å. The snapshot of the structure after compression is presented in Figure 1b along with the single graphene layer chosen from the center of MG. A detailed description of the compression will be described in Section 2.

To study the mechanical properties of the compressed MG, uniaxial tension is applied along all three *x*-, *y*- and *z*-axes at a constant strain rate of $\dot{\varepsilon }$ = 0.005 ps${}^{-1}$ under an NVT ensemble. As well as compression, tensile deformation is applied to the whole simulation box. Stress components are calculated during the simulation. Further compression strain will be defined as $\varepsilon_{cp}$, while tensile strain will be defined as ε. This deformation is prescribed on the simulation box in which periodic boundary conditions are considered. During each time step, the position of each bead is translated from its current position to a new one by a displacement increment that scales with the overall stretch of the simulation box.

## 4. Conclusions

We studied the mechanical response to tensile strain for multilayer graphene through molecular dynamics simulations. The effects of the tension direction and structural morphology on the mechanical properties were analyzed. We demonstrated that compressed multilayer graphene is not isotropic. The structural features obtained as a result of the crumpling of graphene layers under compression play an important role in the mechanical behavior of the graphene network.

This work allows us to understand the process of compression of multilayer graphene with a given distance between layers. One of the methods to obtain such multilayer graphene with a given interlayer distance is the adsorption of metal nanoparticles. This does not affect the mechanical properties but allows us to obtain a larger separation distance than in graphite. The control of the size of metal nanoparticles allows for controlling the value of the interlayer distance.

From the point of view that the main role is the stacking of graphene layers with respect to the tensile direction, several major conclusions can be made:(i)Multilayer graphene shows good mechanical properties under compressive and tensile strain, which can be explained by the strong interaction induced by the crumpling of graphene layers and entangled microstructure.(ii)One of the main deformation mechanisms is the slippage of graphene layers during tension.(iii)Two competitive mechanisms can be distinguished both for compression and tension: the crumpling of graphene layers, which increases the stresses, and the sliding of graphene layers through the surface-to-surface connection, which lowers the deformation stresses.

Such complex graphene architecture can demonstrate outstanding properties—both high strength and exceptional ductility. The present work reveals that such complex crumpled graphene networks can be obtained by controlling the interlayer distance and compression morphology, which can be a step towards the fabrication of graphene structures with better mechanical properties.

## Figures and Tables

**Figure 1 ijms-24-06691-f001:**
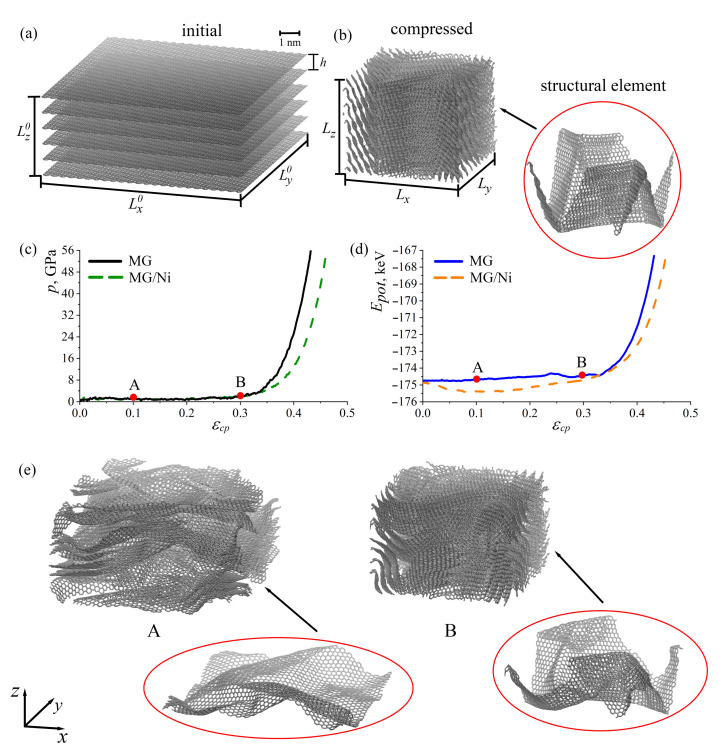
(**a**) Initial structure of multilayer graphene (MG). (**b**) Structure of MG after compression. (**c**) Pressure–strain (**c**) and potential energy–strain (**d**) curves during compression for MG and MG with Ni nanoparticles between layers. (**e**) The snapshots of structure at points: $\varepsilon_{cp}$ = 0.01 (A) and $\varepsilon_{cp}$ = 0.03 (B).

**Figure 2 ijms-24-06691-f002:**
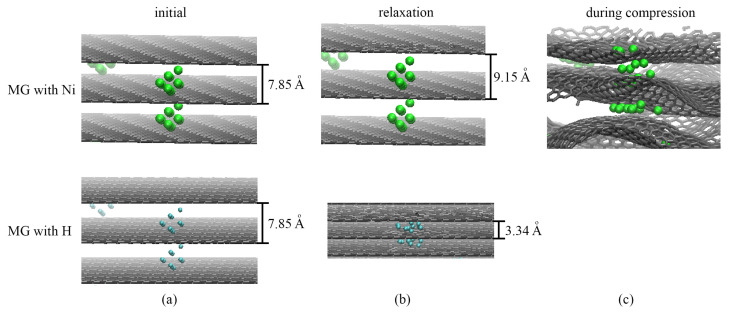
Snapshots of the part of the simulation cell of MG with Ni and H nanoclusters at (**a**) the initial state, (**b**) after relaxation and (**c**) during compression. Nickel atoms are shown in green, H atoms are shown in blue, and the graphene layers are shown in gray.

**Figure 3 ijms-24-06691-f003:**
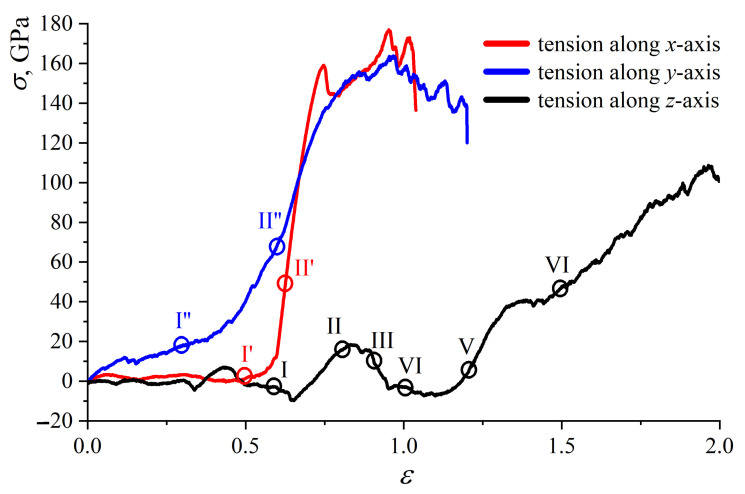
Stress–strain curves during uniaxial tension for MG. Lables I–VI, I’, I”, II’, II” show some critical points for further structural analysis.

**Figure 4 ijms-24-06691-f004:**
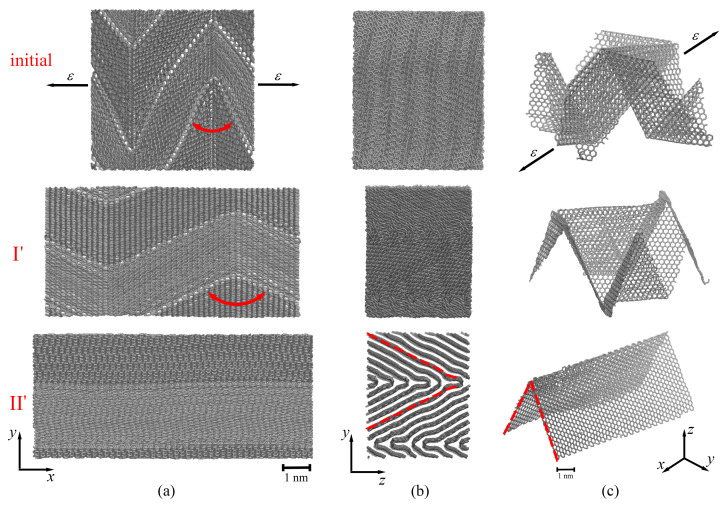
The snapshots of the MG in projection to the (**a**) xy-plane and (**b**) yz-plane and (**c**) snapshot of the graphene layer in perspective view at the initial state and during tension along the *x*-axis at critical points I’ (ε = 0.5), II’ (ε = 0.62). The structure after compression and before tension is labeled as “initial”. Points I’ and II’ are shown in a red solid line in Figure 3. The opening angle is shown by red arrows. The tension direction is shown by black arrows.

**Figure 5 ijms-24-06691-f005:**
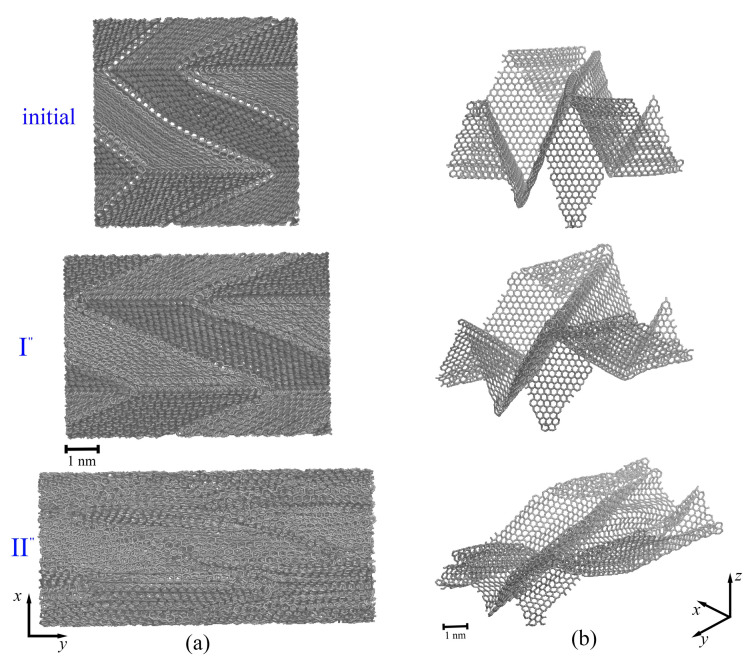
The snapshots of the MG in projection to (**a**) the xy-plane and (**b**) snapshot of the graphene layer in perspective view at the initial state and during tension along the *y*-axis at critical points I” (ε = 0.3) and II” (ε = 0.6). The structure after compression and before tension is labeled as “initial”. Points I” and II” are shown as a blue solid line in Figure 3. The tension direction is shown by black arrows.

**Figure 6 ijms-24-06691-f006:**
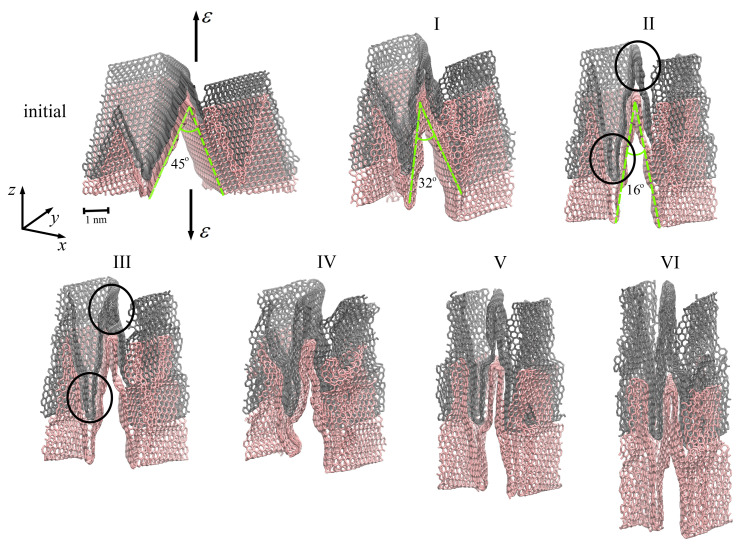
The snapshots of structures at critical points I (ε = 0.56), II (ε = 0.75), III (ε = 0.85), IV (ε = 1.0), V (ε = 1.2) and VI (ε = 1.5) during tension along the *z*-axis. The structure after compression and before tension is labeled as “initial”. Points I–VI are shown as a black solid line in Figure 3. The tension directions are shown by black arrows.

## Data Availability

Data available on request due to restrictions eg privacy or ethical.

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
