# Peer review of "Mechanical Properties of Graphene Networks under Compression: A Molecular Dynamics Simulation"

_ijms, 2023, doi:10.3390/ijms24076691_

Round 1
Reviewer 1 Report
This work investigated the mechanical properties of crumpled multilayer graphene via molecular dynamics simulation. The interlayer distance of multilayer graphene was tuned by Ni or H cluster intercalation. The simulation results of the crumpling process of the graphene multilayer were first analyzed based on the corresponding stress-strain curves and deformation morphologies. The tensile response of the crumpled multilayer graphene was then studied and the effect of loading orientation was analyzed. Overall, this paper is well-written and provides useful information for studies of the mechanical properties of graphene-based nanomaterials and nanostructures, and thus is recommended to be published in the International Journal of Molecular Sciences after the following issues are addressed.
1. In the introduction part, the review work of “recent progress on graphene-analogous 2D nanomaterials: properties, modeling and applications” can be cited.
2. Some simulation details of the application of compressive or tensile deformation should be provided. Generally, there are two ways to apply the deformation. One is by uniformly expanding the whole simulation box while the other is by imposing a constant velocity on the boundaries. Which method was used here? The authors may refer to Chapter 5 of the recent book “Molecular dynamics simulation: Fundamental and applications” for detailed information.
3. Some discussion on the effect of interlayer distance on the tensile behavior of the crumped multilayer graphene may be helpful.
4. There are some grammar errors in the manuscript. For example, on Page 2, Line 47, “attract” should be “attracts”; on Page 6, Line 200, “not appeared” should be “did not appear”, on Page 8, Line 240, “effect” should be “effects”. The authors are suggested to go through the manuscript and minimize such errors.
Author Response
Comment 1
In the introduction part, the review work of “recent progress on graphene-analogous 2D nanomaterials: properties, modeling and applications” can be cited.
Reply:
Thank you for your advice, references [1] have been added to the text.
Comment 2
Some simulation details of the application of compressive or tensile deformation should be provided. Generally, there are two ways to apply the deformation. One is by uniformly expanding the whole simulation box while the other is by imposing a constant velocity on the boundaries. Which method was used here? The authors may refer to Chapter 5 of the recent book “Molecular dynamics simulation: Fundamental and applications” for detailed information.
Reply:
We appreciate the comment, we agree with the reviewer advice. The deformation is applied to the whole simulation box. Text in the “Simulation Details” Section was improved.
Comment 3
Some discussion on the effect of interlayer distance on the tensile behavior of the crumped multilayer graphene may be helpful.
Reply:
Thank you for your advice. The interlayer distance between graphene layers is of high importance for the mechanical properties of such structures. The smaller the interlayer distance, the higher the mechanical properties. Moreover, compactness of the structure can be controlled during its fabrication. Our results were compared with works [41,75] and the changes of the interlayer distance in our structure are additionally analyzed. Additional descriptions were added to the text.
Comment 4
There are some grammar errors in the manuscript. For example, on Page 2, Line 47, “attract” should be “attracts”; on Page 6, Line 200, “not appeared” should be “did not appear”, on Page 8, Line 240, “effect” should be “effects”. The authors are suggested to go through the manuscript and minimize such errors
Reply:
Thank you for such a careful reading of our work, we have corrected grammatical misprints in the text.
Reviewer 2 Report
The manuscript investigated the mechanical properties obtained from compression and tension of multilayered graphene with increased interlayer distance using the MD simulation under biaxial compression and uniaxial tension along three different axes. The tensile strength, ductility and the deformation characteristics due to graphene layers stacking are studied. Some interesting results are obtained that the mechanical properties along different directions are significantly distinguished, and multilayer graphene after biaxial compression can sustain high tensile stresses combined with high plasticity. I believe that the new findings in this manuscript is of interest to researchers in this area. However, these simulation results are not verified by experimental results or simulation results by other papers. Actually, as mentioned by the authors in Introduction, there was a real experiment on the mechanical properties of MG with different interlayer distances [24,25]. The authors can add a comparison between their results and the experimental results in Ref. [24,25] to validate their MD results for supporting their conclusion.
Author Response
Comment 1
The authors can add a comparison between their results and the experimental results in Ref. [24,25] to validate their MD results for supporting their conclusion.
Reply:
Thank you for your advice. We have added some new discussion on the comparison with the experiment.
Reviewer 3 Report
I am sending a review in the attachment.

Author Response
Comment 1
Please specify the title of the journal
Reply:
Thank you for your advice. The journal title is “International Journal of Molecular Sciences”. The journal title was added.
Comment 2
In Figure 1, the potential energy-strain curves during compression for MG and MG with Ni nanoparticles between layers should be labeled Figure 1d. However, Figure 1d as 1e. That will be more visible.
Reply:
Thank you for your advice, we corrected the figures and made changes to the text.
Comment 3
The introduction section should be supplemented with an overview of graphene production - experimental data.
Reply:
We appreciate the comment. To date, more methods for graphene production are known: mechanical and chemical exfoliation of graphite, epitaxial growth of graphene on different substrates, such as silicon carbide, Ni, Cu, Pt, Co etc., organic synthesis, chemical vapor deposition and synthesis of graphene from graphite oxide via oxidation of graphite. We have added some overview (Refs. [19-22]) of graphene production methods to the Introduction.
Comment 4
An overview of the properties of the obtained graphene materials - experimental data should be added to the introduction section.
Reply:
We appreciate the comment. We agree with the reviewer that information about the experimental study of the properties of the obtained graphene materials will improve our manuscript. We have added an overview of the properties of the obtained graphene materials to the Introduction (Refs. [23-32]).
Comment 5
Were the obtained simulation data confronted with the experiment? Have they found application in real systems?
Reply:
Thank you for your advice. We have added some new discussion on the comparison with the experiment.
Comment 6
References: a. 1, 2, 3, 4, 8, 12, 14, 16, 18, 20, 22, 23, 25, 29, 40, 41, 44, 45, 47 - Please enter the page range of the cited article. b. 2, 3 - Please enter the volume of the cited article.
Reply:
Thank you for your advice, page ranges have been added to the text. The manuscripts (Refs. [3,4]) are available online, but do not have a volume or page number yet.
Round 2
Reviewer 2 Report
The authors have added new discussion on the comparison with the experiments.
Reviewer 3 Report
The article can be published in its current form.